# **Evaluating the effectiveness of quantitative descriptions of Earth Science phenomena during outreach activities**

Matteo Trolese<sup>1</sup>, Alessandro Tadini<sup>1</sup>, Laura Pieretti<sup>2</sup>, Damiano Biagini<sup>1</sup>, Spina Cianetti<sup>1</sup>, Simone Colucci<sup>1</sup>, Matteo Cerminara<sup>1</sup>, Claudia D'Oriano<sup>1</sup>, Chiara Montagna<sup>1</sup>, Michele D'Ambrosio<sup>1</sup>, Raffaello Pegna<sup>1</sup>, Giuseppe Re<sup>1</sup>, Francesco Sanseverino<sup>1</sup>, Carlo Giunchi<sup>1</sup>, Carlo Meletti<sup>1</sup>, Tomaso Esposti Ongaro<sup>1</sup>

Correspondence to: Alessandro Tadini (alessandro.tadini@ingv.it), Matteo Trolese (matteo.trolese@ingv.it)

**Abstract.** We present a study aimed at evaluating how experiment-driven communication, in particular in the field of volcanology, seismology and tsunami sciences, is effective in conveying quantitative concepts and in increasing the understanding of natural phenomena. We conducted two dissemination initiatives dedicated to creating general public's awareness and appreciation of geoscience, each targeting a different type of audience. The first initiative was a lesson delivered to high-school students with a humanistic background as part of the European Researchers' Night. The second was an interactive experiment/activity carried out in a booth hosted by the Italian Civil Protection Department, focused on best practices for risk mitigation. It was presented during an international event dedicated to fandom culture (Lucca Comics and Games), which was entirely unrelated to geoscience. The core of both initiatives - focused on volcanic conduit dynamics, earthquake localization and magnitude calculation, and volcanic landslide-induced tsunami – consisted of hands-on experiments, corroborated by the determination of parameter values, adding a quantitative dimension to the phenomenological experience. We also delivered questionnaires to all participants that were aimed at evaluating the effectiveness of the conveyed messages. One questionnaire was delivered to the 62 high-school students during the European Researchers' Night and two questionnaires (pre- and post-experiment) were delivered to 26 participants during Lucca Comics and Games. The results of the survey show that our experiments were well-received and, as a general conclusion, underlined that geophysical hands-on experiences can successfully foster the engagement of the people, even when providing quantitative evidence. However, it is also important to adapt the dissemination initiative to the target audience, by considering the amount of time available and the background of the attendants. For external instructors who present experiments to high school classes, we once again assessed the importance of the synergy between school teachers and external communicators before and after the events, in order to avoid contradictory messages delivered to students. We finally underline the importance of finding new ways to promote a modern and interactive way to communicate geosciences.

<sup>&</sup>lt;sup>1</sup>Istituto Nazionale di Geofisica e Vulcanologia, Sezione di Pisa, Pisa 56125, Italy

<sup>&</sup>lt;sup>2</sup>Istituto Istruzione Superiore Galilei-Pacinotti, Pisa 56125, Italy

#### 30 1 Introduction

disseminate science.

Volcanoes, earthquakes, and tsunamis are natural phenomena that have the power to captivate people of all ages and backgrounds. Concurrently, the hazards posed by such events frequently become points of debate and dialogue between the public and institutional authorities. Increasing the knowledge and awareness of potentially hazardous natural phenomena is an effective way toward community preparedness and risk reduction (Gregg et al., 2004).

Geosciences are often presented to the public in a more descriptive than quantitative way. Many initiatives consist of exhibitions of rocks and minerals, documentary images and videos. Although rock samples and field activities represent the basis of the geological disciplines, and natural phenomena can be explained from a purely theoretical perspective, learning is more effective when a descriptive approach is coupled with practical experiments or unconventional techniques. A recent paper by Jolley et al. (2022) investigated the practices and perceptions of learning, teaching and educational support within volcanology for undergraduate students; most of the interviewed educators used classical tools for teaching, such as rock samples, field activities and experiments. On the other hand, empirical studies indicate that hands-on experiments and computer simulations are effective tools in science education and increase the motivation and interest of students (Rutten et al., 2012; Smetana and Bell, 2012; Winn, 2002; Winn et al. 2006). Hands-on experiments provide the so-called 'situational interest', caused by situation-specific environmental stimulations like novel or attracting activities (Lin et al., 2013), but they often simplify the real complexity to focus on specific aspects. Computer simulations are more suitable for communicating complexity, but often lead to a high 'cognitive load' (i.e., cognitive capacities of learners are overburdened), potentially hindering the learning process (De Jong, 2010; De Jong and van Joolingen, 1998). Kruger (2021) shows that a combination of hands-on experiments and computer simulation represents an efficient learning tool to obtain the best advantages in achieving a comprehensive understanding and the necessary interest for active participation. Hands-on experiments should be at the same time informative, relatively simple to realise, and effective in communicating and engaging participants. This is not trivial for natural phenomena characterized by multiple spatial and temporal scales, such as volcanic eruptions and earthquakes, as they are not easily adapted to in-classroom experiments or demonstrations. Adapting the wide range of natural phenomena to the laboratory environment requires a dimensional scaling of the measured quantities (Merle, 2015). To become more effective, this aspect needs to be properly addressed during outreach activities. One recent example is from Wadsworth et al. (2018), who developed "trashcano", an experiment based on the experience of Harpp et al. (2005) in which an explosion is caused by the rapid expansion of over-pressurized gas, driving in turn the acceleration of particles such as table tennis balls. Similarly, Moutinho et al. (2016) proposed an experiment showing the effect of the 1755 earthquake in Lisbon (Portugal). This experiment was proposed to 126 high-school students and simulated the seismic effects on buildings as a function of their distance to the epicentre and to the presence of different rock layers with different physical properties. Results of this experience allowed the authors to conclude that model-based learning is an important methodology to

The aim of our work is to evaluate how public understanding of complex natural phenomena can be enhanced by dissemination activities that include hands-on experiences involving quantitative approaches and descriptions. We show simple but illustrative experiments where the public is expected to contribute actively to measurements and their discussion, and comment on their perception and effectiveness. Our work highlights how tailored, interactive experiments combined with computer simulations can enhance public understanding of geophysical processes while addressing broader objectives in both STEM (Science, Technology, Engineering, and Mathematics) education and disaster resilience, ultimately contributing to bridge the gap between geoscience research and societal impact. This paper discusses the effectiveness of public outreach initiatives in the fields of volcanology and seismology, performed by researchers from Istituto Nazionale di Geofisica e Vulcanologia (INGV - National Institute of Geophysics and Volcanology, a public research institute under the Italian Ministry of University and Research) in Pisa, Italy. The activities described were implemented by the INGV-Section of Pisa and targeted two different audiences, high-school students and the general public, in two different dissemination activities. Researchers developed innovative hands-on experiments that enabled discussion of key concepts. Both events incorporated pre- and post-engagement assessment to evaluate the participants' understanding, reflecting a growing emphasis on mixed methods evaluation in science communication. By integrating interdisciplinary approaches such as physical experiments, computer simulations, participatory pedagogy, and social science methodologies, this study contributes to geoscience communication goals of advancing robust, inclusive geoscience engagement. This paper is organized as follows: Section 2 provides an overview of the dissemination initiatives. Sections 3 and 4 offer, respectively, technical details regarding the implementation of the experiments and simulations and an analysis of the questionnaire motivation and results obtained from both audiences. Section 5 discusses the challenges encountered during the experiments and outlines key take-home messages for enhancing communication effectiveness. Finally, Section 6 presents our concluding considerations.

#### 2 Dissemination initiatives

INGV has a long-running experience in divulgating geo-scientific knowledge, demonstrated by multiple dissemination events (see e.g. D'Addezio et al. 2014; Riposati et al. 2020; Cianetti et al., 2021; D'Addezio 2025). In this work, we report two initiatives within the framework of two national events, the European Researchers' Night - BRIGHT-NIGHT (https://bright-night.it/) and the "Io non rischio" campaign (https://iononrischio.protezionecivile.it/en/know/campaign-manifesto/).

The two initiatives described in this paper reflect different approaches needed to deal with two different contexts. In the first case, researchers organized the activity over two hours with a high school class. In the second case, they operated within a public event and had only a few minutes to capture the attention of people of various ages and backgrounds. While in the

first case there was time to delve deeper into the experiments and actively involve the students, in the second case it was necessary to come up with something eye-catching and engaging to attract the attention of passers-by.

For both initiatives, questionnaires were administered to participants to evaluate the effectiveness of the geoscience communication activities. All the participants were informed of the questionnaires' purpose and guaranteed full anonymity.

Figure 1: a-d) pictures taken during the BRIGHT-NIGHT and e) Io non rischio ("I don't take risks") initiatives showing the engagement of the students and the general audience.

# 2.1 BRIGHT-NIGHT







BRIGHT-NIGHT (Fig. 1a-d) is one of the projects of the European Researchers' Night (https://marie-sklodowska-curie-actions.ec.europa.eu/actions/msca-citizens/join-a-celebration-of-science), funded by Marie Skłodowska-Curie Actions, an initiative conceived by the European Commission with the goal of spreading scientific culture and showcasing the social impact of research. BRIGHT-NIGHT - "Brilliant Researchers Impact on Growth Health and Trust in research night" - is a partnership among universities, national research institutions, and other research organizations across Tuscany, Italy. Among the outreach activities proposed within the BRIGHT-NIGHT project, some are aimed at the general audience, while others are dedicated to students from primary to high school who have the opportunity to visit the facilities of the hosting scientific institutions. BRIGHT-NIGHT is a unique opportunity to raise awareness and engage the general public, especially young students, in the world of Earth Sciences.

Within the BRIGHT-NIGHT framework, we conducted an in-person lesson for high-school students in Pisa (specifically, class 4C from "Liceo Classico G. Galilei"). In this type of high school, humanistic disciplines form the core of the curriculum, while scientific subjects represent a smaller part. The lesson was designed to be interactive, allowing students to participate actively by performing experimental measurements. The class was split into two groups that participated both in volcanological and seismological experiments.

At the end of the lesson, the students were asked to complete a questionnaire designed to evaluate both the effectiveness of the conveyed messages and their understanding of Earth and physical processes. The questionnaire was administered not only to the class that participated in the experiments (class 4C) but also to three other classes (4A, 4B and 4D), of the same age and from the same school, which did not attend the lesson. This allowed us to evaluate the perception of the two audience groups with respect to the same type of information. All classes had previously attended in-person lessons at their institution on the topics related to the experiments, which provided them with a common background.

#### 2.2 Io Non Rischio

"Io non rischio" (Fig. 1e; "I don't take risks") is a public communication campaign focused on good practices for risk mitigation, organized by the Italian Civil Protection Department. It combines the efforts of science, volunteering associations and institutions at national and local levels to turn awareness into action throughout the years.

Objectives of this initiative are: 1) raising awareness of natural and human-caused hazards affecting individuals and communities; 2) encouraging preventive actions and good behaviors to mitigate risks; and 3) fostering a culture of civil protection by teaching people what to do before, during, and after emergencies.

Many events are organized every year; among them, the Civil Protection Department and INGV managed a booth about civil protection best practices at the 2024 edition of the world-renowned Lucca Comics and Games festival (https://www.luccacomicsandgames.com/i-festival/lucca-comics-games/) in Lucca (Italy). INGV Pisa participated with an experiment aimed at showing the potential hazard associated with tsunamis, specifically with an analogue experiment simulating tsunami wave propagation. During this experience, visitors were able to observe and measure the waveform of a landslide-triggered tsunami and were invited to complete two online questionnaires (administered before and after the experiment), which were partially inspired by those of Amato et al. (2024).

#### 3 Material and Methods




#### 3.1 Design of hands-on experiments

# 3.1.1 Viscosity and bubbles: the physics of volcanic eruptions

The experimental setup (Fig. 2) was designed to simulate the ascent of gas bubbles in liquids with different viscosities. The aim was to illustrate the role of viscosity in controlling the rise of gas bubbles in volcanic systems and its influence on the eruptive style (explosive vs. effusive). The hands-on experiment was complemented by a practical exercise in which students measured the ascent speeds of the bubbles using real-time video imaging. By balancing the equations of the relevant acting forces, the students used the acquired data to estimate the viscosity of the liquid. This hands-on experience demonstrates how researchers collect and process data, using modern instrumentation to infer otherwise inaccessible process properties. Finally, we presented the results of a numerical simulation, reproduced using the open-source CFD library OpenFOAM, which reproduces the experimental setup (Fig. 3b). A movie of the simulation is provided in Video Supplement 1. Computational volcanology is one of the pillars of scientific research developed at the Pisa section of INGV, and the activity aims to communicate the importance of numerical simulations in volcanological research. By adding a quantitative aspect to this experiment, we aimed to highlight the technological applications in modern volcanology and promote core STEM skills. As shown in Figure 2, the experiment setup consists of two cylindrical containers made of transparent synthetic glass, each filled with a Newtonian fluid of a specific viscosity. These fluids represent two end-members of a viscosity spectrum, ranging from very low (water) to very high (golden syrup). Some experiments used molten real magma or silica glasses at temperatures ≥ 900 − 1 100 °C to illustrate the dynamics of volcanic processes at a general public (Wadsworth et al., 2019).

In the context of our purpose, we used an analogue, low temperature material (see also Rust et al., 2008), which is easily and safely transported outside the laboratory, allowing us to quantify the physical properties being transparent.

A schematic representation of the apparatus is shown in Fig. 2a and consists of:






- An air injection system (at the base of each container) composed of a syringe (60 ml) fitted with a Luer-lock connector. Connected to the syringe is a 1.5 m long tube with a diameter of 0.4 mm and a nozzle. The Luer-lock valve facilitates a secure connection, ensuring that once the valve is closed, the pressure of the liquid column does not force the entrance of liquid into the tube.
- A 0.6 m high container with a diameter of 10 cm filled with a test fluid.
- A 25 fps webcam mounted on a vertical support to capture the bubble motion.

The experiment (Fig. 2b) consists of injecting air at the base of the apparatus, which leads to bubble formation and subsequent rise through the fluid (with the ascent dynamics influenced by the fluid viscosity). The rising bubbles are recorded by the webcam, and the acquired images are processed in real time to extract the bubble position and size over time. These parameters are then plotted to determine the radius and ascent velocity of the bubbles (Fig. 3a).

To process the video images and extract quantitative data, a Python-based algorithm was implemented. The algorithm utilizes a background subtraction method to isolate moving objects (bubbles) from a static background. Once the contours are extracted, the centroid of each bubble is computed from the bounding rectangle, allowing the algorithm to track bubble motion between consecutive video frames. For velocity calculation, the algorithm compares the vertical centroid positions from successive frames. The change in position is divided by the time interval between frames to compute the bubble rise velocity in pixel units. Using a conversion factor, the velocity is then expressed in cm/s. This tracking process is repeated for each frame, and the resulting data points are used to plot the bubble position and compute a best-fit line through a moving window of points. The slope of this line corresponds to the bubble rising velocity (Fig. 3a).

The experiment is designed so that high-school students can directly take measurements of the bubble radius (R) and its rise velocity (u). With known values for the liquid and gas densities ( $\rho_l$  and  $\rho_b$ , respectively) and gravitational acceleration (g), students calculate the viscosity of the liquid ( $\mu$ ) by applying the following formula (Clift et al., 1978), derived from the balance of Archimedes' gravitational and viscous forces:

$$\mu = \frac{\rho_l - \rho_b}{u} \frac{2}{9} R^2 g$$

Since the viscosity of water is relatively well established (approximately 10<sup>-3</sup> Pa/s), measurements are performed primarily for bubbles rising in the golden syrup, whose viscosity varies with the sugar-water concentration (Schellart, 2011). The

calculated viscosity value is then used as input for the numerical simulation. Finally, the numerical results are shown to confirm that the bubble ascent times in the simulation are consistent with those observed experimentally (Figure 3b).

Figure 2: a) scheme of the bubble apparatus for both the water- and golden syrup-filled cylinders; b) a picture of the bubble 190 experiment showing the bubble rise. Bubble size is not due to decompression, but rather to differences in the initially injected volume.

Figure 3: Analysis of rising bubble dynamics. a) Left - single video frame at the indicated time (shown in the upper left), with the detected bubble outlined. Overlaid annotations report the instantaneous bubble radius (in cm), determined from half the bounding-box width, and the instantaneous rise velocity (in cm/s), computed from the vertical displacement between consecutive frames. Right - space-vs-time plot of the bubble trajectory over the most recent 20 s. Points mark the measured heights (in cm) at each video frame; solid black lines are linear fits computed over five consecutive video frames. The slope of each fit corresponds to the average rise velocity over that five-frame interval. b) numerical simulation of the experiment. Simulations are performed using the open-source computational fluid dynamics software OpenFOAM® (v2306), which has been tested and benchmarked on problems involving bubbles (Brogi et al., 2022; Colucci et al, 2024). The full video is available as Video Supplement 1.

#### 3.1.2 Discovering Earthquakes: epicenters, hypocenters, and the power of the Earth





The primary goal of the seismological laboratory is to educate students about the fundamental concepts of epicenter, hypocenter, and magnitude, thereby fostering a deeper understanding of how earthquakes occur and are analyzed. To achieve this, an interactive and illustrative workshop was conducted, guiding participants through the basic methods used to quickly estimate the main parameters of an earthquake. This approach helps to make some complex topics simpler and more relatable, from the basic meaning of scientific terms to the actual methods scientists use, which we often hear about in the news during earthquake events.

During the activity, researchers walk students through the entire workflow of earthquake monitoring — from the functioning of seismic instruments to the interpretation of seismograms — making complex scientific processes accessible and engaging. The lesson begins with a brief introduction to plate tectonics and different types of faults, accompanied by examples of visible surface effects on the ground caused by local faults following recent major earthquakes in Italy. This is followed by an explanation of the concepts of epicenter and hypocenter, highlighting their differences. An in-depth explanation is provided on body and surface seismic waves and their propagation through different materials, such as solids and liquids. The way body seismic waves propagate through the Earth provides key insights into the structure of its interior.

Once the fundamental concepts were introduced, the hands-on activity started, featuring real-time waveform displays from a seismometer. To explain how a seismic station operates, the researchers use a Raspberry Shake RS3D seismometer (see Fig. 4), equipped with three 4.5 Hz geophones orthogonally oriented along the vertical, north-south and east-west axes. The system also includes a Raspberry Pi 3 Model B that records, digitalizes and stores data at a sampling rate of 100 samples per second (Fig. 4).

Figure 4. a) Raspberry Shake RS3D seismometer; b) plot of waveforms and the frequency spectra for the three components on the right; c-d) pictures of the exercise to determine the location and magnitude of an earthquake.



Students are then encouraged to jump in place to observe how the seismic station records the ground motion generated by their activity. This demonstration allows them to clearly distinguish between background noise and an actual signal, and provides them with the tools to identify the onset time of seismic waves on a seismogram. This experience also allows students to familiarize themselves with two additional key concepts. The first, which is essential for calculating the earthquake's Richter magnitude, is that the greater the energy released by the seismic event, the larger the amplitude of the

shear waves recorded on the seismogram. The second concept highlights the extreme sensitivity of seismic instruments. To minimize interference from non-seismic sources, which are considered noise in seismic monitoring service, these instruments must be installed in locations free from human activity. Once students feel more comfortable reading the seismic signals they are asked to determine where an earthquake happened and how strong it was.

The students are provided with five seismograms recorded corresponding to an earthquake that occurred in central Apennines; the travel-time curves for the arrival of P and S waves; a geographical map showing the locations of the seismic stations. The aim of the exercise is to determine both the location and the Richter magnitude of the earthquake using a ruler and a compass. (see Fig. 4)

The activity is split into four steps:



- 1. **Explaining waveform data and significant parameters.** Researchers guide the students through the interpretation of the seismic signals and highlight the importance of accurately determining the onset time of P- and S-phases: due to the high values of P and S wave velocities, even a relatively small uncertainty can lead to elevated error of the epicentre.
- 2. Determining P- and S-phases onset time and figuring out the time difference between them (ΔT = ts-tp). Students learn how to pick the different types of seismic waves and compute the interval between their arrival times. Using the travel-time curves, ΔT (s) is converted into distance ΔS (km) between source and receiver. The travel-time is indeed the equation that describes the time needed by the seismic wave to travel from the source to the receiver.
- 250 3. Locating the earthquake's epicentre. Using a compass and taking into account map scale, the students draw five circles on the map centred in the position of each station and with radius corresponding to the distances previously determined. The intersection of the circles on the map is the epicentre of the earthquake.
  - 4. Calculating the Richter Magnitude (ML). Students are also requested to measure the amplitudes of the S wave in the two horizontal directions (North-South and East-West) of the seismograms (called 'amplitude' or 'amp'). The corresponding values are used with the ΔS previously determined to evaluate, for each direction, ML as:

$$ML = log(amp) + 1.110 * log(\Delta S) + 0.00189 * \Delta S + 3.591$$

The magnitude of the earthquake is finally computed as the average of all the ML calculated at each station.

Although we used a simplified method with respect to what is done in the routine monitoring service, and the quite large human error in measuring the amplitude on paper, the epicentral parameters determined for this earthquake are very close to those provided by the INGV Bulletin.

# 3.1.3 Not just simple waves: tsunami hazard at volcanic islands




This experiment was designed to demonstrate to participants how a tsunami propagates, what are its most relevant measurable parameters (wave form and amplitude), and how such measurements can be useful to implement an early-warning system. In parallel, supporting videos of real tsunamis or large-scale experiments are shown to the public to provide additional information on tsunami effects on people and buildings.

For the experiment (Figure 5), we developed a simple apparatus to model both tsunami generation and propagation. The setup is composed of a horizontal, transparent channel with a rectangular-cross section, measuring 1.520 m in length, 0.215 m in height, and 0.093 m width, and is filled with still water. On one end of the channel, an inclined plane measuring 0.295 m in length and 0.145 m in height with a slope angle of  $\theta = \sim 26^{\circ}$  is partially immersed in the water. Tsunami waves are generated when a rigid block, initially placed on top of the inclined plane, is released and slides along the surface.

To capture the tsunami waveform, a water level sensor is positioned approximately 0.8 m from the end of the channel containing the inclined plane. This sensor measures the water elevation in the channel. It is a resistive sensor consisting of four pairs of tin-plated copper wires, spaced 3 mm apart and 6 cm long, positioned vertically and partially immersed in water. It measures the electrical resistance offered by the water on the immersed portion of the wires. As part of a resistive voltage divider, the sensor output is sampled at 100 Hz by an Arduino microprocessor via its ADC. The measured voltage is then converted into a wave height and transmitted in real time via a virtual USB serial connection to a personal computer for graphical visualization.

At the start of the experiment, the rigid block slides from a subaerial position along the inclined plane, generating a tsunami wave that propagates along the apparatus. When the first wave reaches the sensor, its waveform is captured and displayed in real time. Additionally, once the wave amplitude exceeds a predefined threshold (3 cm in our setup), an alarm is triggered. This alarm reproduces the Super Mario theme, simulating an operational tsunami alert system while also leveraging the popularity of the tune to attract attention (especially in light of the adjacent Nintendo stand).

This experiment simulates a scenario in which a subaerial landslide enters the sea. Alternatively, the experiment can be modified so that the rigid block is moved rapidly from an underwater resting position to a subaerial position, thereby modeling the generation of a tsunami triggered by a submarine landslide.

Figure 5: a) scheme of the tsunami apparatus showing the rectangular box filled by water, and the different components of the apparatus (inclined plane, rigid block and sensor); b) picture of tsunami experiment.

# 3.2 Questionnaires


We developed classical style questionnaires with true/false or multiple-choice questions, rather than Likert-scale type (as for example in O'Connor et al., 2023). This was motivated by two considerations. First, we aimed at having questions with a lower degree of complexity than choosing an answer on a scale 1-5. Indeed, Likert-type scales require respondents to

translate an 'agreement' into a numerical level, thereby increasing cognitive load (Tourangeu et al., 2000; Bradburn et al., 2004). Direct response items (true/false, classic multiple choice) are cognitively easier and less prone to variable interpretations by respondents. Second, classical-like questionnaires are more appropriate for small groups of participants and/or with diverse backgrounds, as in our case (see e.g. Presser and Blair, 1994; DeVellis and Thorpe, 2021). The questionnaires are provided in both Italian and English as supplementary material.

Each question was designed to evaluate the learning of the audience about a topic that was explored in selected experiments. We used close-ended questions, including questions requiring "yes" or "no" answers (e.g., "would you like to work in a scientific field in the future?"). We did not link pre- with post-assessment questionnaires to guarantee full anonymity.

We asked the participants to use their mobile phones to access the questionnaire implemented on Google web form via a QR code, ensuring accessibility across various devices and operating systems. For the students only, the Google form was made available by the link sent by the teacher to the WhatsApp group comprising all the students in the class. The questionnaire required approximately 20 minutes to be completed.

# 4 Results of questionnaires

#### 310 4.1 School questionnaires





classes that did not attend the lesson (4A - 4B - 4D). The questionnaire was organized into two sections: the first contained general questions (Figs. 6-7) while the second focused on more specific topics addressed with the experiments (Figs. 8-9). Comparing the answers on general topics from the two groups (Fig. 6), we found that most students reported slight to moderate interest in scientific research: in group 4C, 20% were slightly interested, ~67% moderately interested, and ~13% very interested; in groups 4A–4B–4D, 34% were slightly interested, ~49% moderately interested, and ~11% very interested (Q1b). Also, when asked about their interest in pursuing a scientific career, the two groups provided analogous answers: ~40% of students expressed no interest, and ~47% responded with "maybe" (Q2b). The percentage changed when students were asked if they could be interested in studying Earth Science or geophysical disciplines: while ~62% of students in classes 4A-4B-4D stated that they were not at all interested, ~73% of the students from class 4C reported being slightly interested (Q3b). As expected, INGV as a public research institute was much better known among class 4C respondents, with ~67% being able to explain the institute in detail compared to only ~4% in the other groups (Q4b). An important aspect is related to Q5b, which investigated the importance of practical activities. Interestingly, 80% of the students in class 4C believe that practical or laboratory activities are essential for understanding natural phenomena, while only ~60% of the students who did not attend the lesson shared this view, emphasising the effectiveness of our approach.

A total of 62 students completed the questionnaire; 14 from the class that participated in the laboratory (4C) and 48 from

Concerning the questions on acquired knowledge (Fig. 7, Q7b-Q10b), over 90% of the students in class 4C reported that they understood the concepts covered in the dissemination activity to a moderately or high degree, compared to 80% - 90% for students in classes 4A-B-D.

Overall, the responses to the specific questions (Fig. 8-9) indicate that participation in the initiative led to an increase in knowledge on certain topics. Concerning the volcanology-related questions (Q11b-Q15b), the students in class 4C performed better on questions O11b-O12b-O13b-O15b. Approximately 73% of these students answered correctly to the question "Which factors influence the type of an eruption?" (Q11b), in comparison to ~66% of students from the other classes, while both groups achieved similar success on the question "What is the role of viscosity in explosive eruptions?" (Q12b) with 335 ~73% correct responses. For O13b (relationship between magma viscosity and rising speed of gas), ~80% of class 4C answered correctly, versus ~64% in classes 4A-B-D, and for O15b (importance of combining numerical and laboratory experiments), ~87% of class 4C responded correctly compared to ~83% in the other groups. However, O14b (calculation of the viscosity of a liquid) was answered more accurately by students from classes 4A-B-D (~85% correct) than by those from class 4C (~73% correct). Regarding seismological-related questions (Q16b-Q20b), the data reveal some interesting trends. 340 For Q16b (earthquake magnitude increase from M3 to M5), students in class 4C performed better with ~47% correct answers compared to ~28% in the other groups, whereas for O17b (find the correct answer about seismic waves) and O18b ("What are dromocrones?") both groups obtained similar results, with correct responses rates of ~80% on each question. O19b ("What does a seismometer measures?"), however, was a challenge for both groups, with only ~20% of class 4C and ~28% of classes 4A-B-D answering correctly. Finally, for Q20b (differences between Mercalli scale and Richter scale), students 345 from classes 4A-B-D provided a higher percentage of correct answers (~60%) than those from class 4C (~40%).

Figure 6: results questionnaire for generic questions Q1b-Q5b for a) the class who participated in the lesson (4C) and b) the classes who did not participate in the lesson (4A-4B-4D).

Figure 7: results questionnaire for generic questions Q6b-Q10b for a) the class who participated in the lesson (4C) and b) the classes who did not participate in the lesson (4A-4B-4D).

Figure 8: results questionnaire for specific questions Q11b-Q15b for a) the class who participated in the lesson (4C) and b) the classes who did not participate in the lesson (4A-4B-4D). Correct answers are highlighted in bold.

Figure 9: results questionnaire for specific questions Q16b-Q20b for a) the class who participated in the lesson (4C) and b) the classes who did not participate in the lesson (4A-4B-4D). Correct answers are highlighted in bold.

#### **4.2 General audience questionnaires**

Twenty-six participants completed both pre- and post-experiment questionnaires. The pre-experiment questionnaire indicates that the participants already had a good understanding of the tsunami phenomenon, as reflected in questions Q2i-Q6i (Fig.10). In particular, most of the participants were aware of the potential tsunami risks of the locations they visited (~69% for Q3i), they understood both the processes behind tsunami generation and the potential impacts on people (~73% for Q4i ~88% for Q5i), and they had a reassuring confidence in scientific investigation (~70% - Q6i). In contrast, given that the majority of participants live >3 km away from the coast (~85% - Q1i), their perception of the tsunami hazard in their own country is less developed (~73% of them answered that their country is not at tsunami risk - Q2i). This likely reflects the fact that, as all respondents were Italian, they did not perceive Italy as a country at high risk for tsunamis, thereby underestimating its potential impact.

370


Concerning the post-experiment questionnaire, in addition to the general appropriateness of both the participants' competences in understanding what is shown in the experiments (Q7i) and the language used by the demonstrators (Q8i), the experiment stimulated the interest in research of most of the respondents (Q9i). This is further supported by the results in Q10i-Q12i, where the majority of respondents answered correctly:


- for Q10i ("On average, at what speed do tsunami waves travel at Stromboli?"), > 80% of the respondents answered correctly.
- for Q11i ("What information can we directly obtain from the study of waveforms?"), > 40% of the respondents answered correctly.


• for Q12i ("Which geographical areas of Italy do you think are most likely to be affected by volcanically-induced tsunamis?"), > 75% of the respondents answered correctly.

We highlight that responses to Q10i-Q11i demonstrate that even complex and quantitative scientific contents were well received, while Q12i testifies that most participants gained fundamental knowledge about tsunami hazard in Italy, information they lacked beforehand (see Q2i).

Figure 10: results of the pre-experiment questionnaire for the tsunami experiment. Correct answers (Q4i-Q6i) are highlighted in bold.

Figure 11: results of the post-experiment questionnaire for the tsunami experiment. Correct answers (Q10i-Q12i) are highlighted in bold.

#### 5 Discussion







One of our primary objectives was to demonstrate that the general public can be successfully engaged in quantitative, and not only in descriptive, geo-scientific dissemination activities (e.g., Ma and Zhang, 2019). Indeed, digital technologies are revolutionizing how geology and science in general are conducted (e.g., Darnila et al., 2018; Pollyea et al., 2018; Ripepe and Lacanna, 2024; She et al., 2022; Zhao and Chen, 2021). Yet, to keep pace with this technological evolution, geoscientists are being encouraged to more effectively convey their technical expertise to non-technical audiences through effective communication (Illingworth et al., 2018; Illingworth, 2023).

Results from our questionnaires (Section 4) indicate that the proposed experiments triggered the active involvement of the audience, enhancing their understanding of the observed phenomena. We acknowledge that the sample size in terms of respondents was small (on the order of 10s); nevertheless, we believe that the results are meaningful, especially given the close-ended rather than Likert-style questions we chose for the surveys.

The two dissemination initiatives were conducted in different contexts, with different audiences, and therefore, we exploited different dissemination strategies.

# 5.1 High school students

The students came to follow a dedicated lesson and had been (at least partially) prepared by their teachers. In this case, combining analog experiments, physico-mathematical reasoning, computer simulations and lessons was a good strategy to obtain the best advantages in achieving a comprehensive understanding of the phenomena and capturing the interest of the students. To prepare the lesson, we took into account the following: i) although students were already accustomed to frontal lesson, it was important for them to not perceive the dissemination initiative as a 'classical' lesson with passive listening, but rather a participatory experience in which each student had to produce and analyze the result of an experiment; ii) the students have a main focus on humanistic subjects and their potential interest for the future is therefore oriented in such direction, while our effort was also to show them that their capability to analyze the problems and abstract them (something they should be already familiar with) is important also in scientific disciplines iii) it was important for us not to take any concept (even the basic ones) as granted.

Survey responses show that the class that participated in the activity understood that experiments are an important part of the scientific method but should always be complemented by field observations and simulations. On the contrary, a significant number of students from the classes who did not participate do not show the same awareness. For the scientific contents, the message was generally well received with a few exceptions. Particularly, for Q19b and Q20b (Fig. 9), incorrect answers given by the students are due to potentially contradictory messages given by the schoolteachers and the INGV

communicators: this highlights the importance of a clear coordination between school and external communicators, before and after the events.

The questionnaire responses also show that INGV as a public research institute is not very popular among students, and possibly the general public: only those who attended our lesson indicated they knew it. We expect this to be specific to areas, like Tuscany, where occurrences of earthquakes are rare and there are no active volcanoes; INGV is indeed very popular in hazard-prone areas such as Naples, Sicily or the Apennines (Pignone et al., 2022). To overcome this limitation, we need to do better to inform people of our mission, through such kinds of activities, and social media.

# 430 **5.2 General public**




People participating in the tsunami experiment did not come specifically to see our dissemination initiative, but rather to attend a completely different event (i.e. Lucca Comics) and that eventually stopped by our stand. Participants to the event came from all over the world, but only the Italian public was at least partially aware of the "Io non rischio" event. Our challenge was therefore to attract the attention of people immersed in worlds of fandom, considering that they potentially had very limited time to dedicate to our initiative. Therefore, we had to focus on conveying the main message about tsunami hazards only. Our strategy has been tailored to the specific context of Lucca Comics: we offered participants a playful and entertaining experience that enabled us to attract the public to the stand, despite the surrounding attractions. We found that whenever interest began to wane, deploying our custom-built, Super Mario-themed device rekindled engagement.

Our results show that the activity about tsunami awareness was generally well received. All respondents absorbed the concepts and answered correctly, but, more importantly, the vast majority changed their perception about scientific knowledge (see Q13i, Fig. 11).

# **6 Conclusions**

Dissemination of scientific culture is crucial for a critical-thinking and healthy society, especially in these historical times, overwhelmed by misinformation and declining trust in science (Budak et al., 2024; Cologna et al., 2025; Larson and Bersoff, 2025). By stimulating deductive reasoning mechanisms and capabilities, quantitative experiments and multidisciplinary activities can contribute to increase the awareness of natural phenomena and their consequences. We have demonstrated how people from different backgrounds can be directly engaged in scientific dissemination, including quantitative aspects. These activities, although complex, can indeed be proposed to a large audience without any prejudice regarding their perceived difficulty. Filling out the surveys possibly represented a further motivation for the audiences to elaborate and cement the information provided. Dissemination needs to be tailored to the target audiences, including engagement through games, collective experiments, and measurements.

# Data availability

Results supporting the findings of this study are presented in the paper.


#### Video supplement

This paper contains a Video Supplement (Video Supplement 1).

#### **Author contribution**

Conceptualization: AT, MT, TEO, CM, CdO, SpC. Methodology: MT, AT, DB, MdA, CG, CM, RP, SpC. Software: MT, MC, SiC, SpC, FS. Investigation: all authors. Data Curation: AT, MT, DB, MC, SpC, SiC, CdO, CM, LP. Writing - Original Draft: AT, MT, DB, SpC, MdA, RP. Writing - Review and editing: all authors. Visualization: AT, MT, DB, MdA, SiC, FS.

# 465 Competing interests

The authors declare that they have no competing interest.

# **Ethical statement**

All participants in this research were fully informed about the study's purpose and procedures. They provided their consent to participate in accordance with ethical guidelines. No identifying information was retained or associated with the responses. Participants were informed that they could choose not to answer any question that they felt uncomfortable with or that could potentially offend them in any way. As for the class experiments, the study was conducted with the head of the school consent, adhering to ethical guidelines. No ethical committee approval is required for the type of information collected.




#### Acknowledgements

We would like to thank all the participants of INGV Pisa section to the initiatives described in this paper. We also acknowledge the help of Lorenzo Cugliari from INGV-ONT (Rome) for insightful suggestions during the preparation of the questionnaires for the "Io Non Rischio" initiative. We thank two anonymous reviewers for insightful comments that improved the quality of the manuscript. The editorial handling of Lewis Alcott is also greatly acknowledged.

#### Financial support

This project has been conducted in the framework of institutional INGV activities for dissemination and outreach. The BRIGHT-NIGHT initiative was supported by INGV and by University of Pisa. The "Io non rischio" initiative is financially supported by the Italian Department of Civil Protection, and by INGV.

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
