# Peer review of "Evaluating the effectiveness of quantitative descriptions of Earth Science phenomena during outreach activities"

_EGUsphere, 2025_

## Referee Comment (RC2)

**Evaluating the effectiveness of quantitative descriptions of Earth Science phenomena during outreach activities**

Matteo Trolese[1], Alessandro Tadini[1], Laura Pieretti[2], Damiano Biagini[1], Spina Cianetti[1], Simone Colucci[1], Matteo Cerminara[1], Claudia D'Oriano[1], Chiara Montagna[1], Michele D'Ambrosio[1], Raffaello Pegna[1], Giuseppe Re[1], Francesco Sanseverino[1], Carlo Giunchi[1], Carlo Meletti[1], Tomaso Esposti Ongaro[1]

[1]Istituto Nazionale di Geofisica e Vulcanologia, Sezione di Pisa, Pisa 56125, Italy
[2]Istituto Istruzione Superiore Galilei-Pacinotti, Pisa 56125, Italy

[revised manuscript text omitted]

Overall, the responses to the specific questions (Fig. 8-9) indicate that participation in the initiative led to an increase in knowledge on certain topics. Concerning the volcanology-related questions (Q11b-Q15b), the students in class 4C performed better on questions Q11b-Q12b-Q13b-Q15b. Approximately 73% of these students answered Q11b correctly, in comparison to ~66% of students from the other classes, while both groups achieved similar success on Q12b with ~73% correct responses. For Q13b, ~80% of class 4C answered correctly, versus ~64% in classes 4A-B-D, and for Q15b, ~87% of class

4C responded correctly compared to ~83% in the other groups. However, Q14b was answered more accurately by students from classes 4A-B-D (~85% correct) than by those from class 4C (~73% correct). Regarding seismological-related questions (Q16b-Q20b), the data reveal some interesting trends. For Q16b, students in class 4C performed better with ~47% correct answers compared to ~28% in the other groups, whereas for Q17b and Q18b both groups obtained similar results, with correct responses rates of ~80% on each question. Q19b, however, was a challenge for both groups, with only ~20% of class

4C and ~28% of classes 4A-B-D answering correctly. Finally, for Q20b, students from classes 4A-B-D provided a higher percentage of correct answers (~60%) than those from class 4C (~40%).

[Figure]

**Figure 6: results questionnaire for generic questions Q1b-Q5b for a) the class who participated in the lesson (4Cs) and b) the classes who did not participate in the lesson (4A-4B-4D).**

[Figure]

[Figure]

**Figure 7: results questionnaire for generic questions Q6b-Q10b for a) the class who participated in the lesson (4C) and b) the classes who did not participate in the lesson (4A-4B-4D).**

[Figure]

[Figure]

[Figure]

**Figure 8: results questionnaire for specific questions Q11b-Q15b for a) the class who participated in the lesson (4C) and b) the classes who did not participate in the lesson (4A-4B-4D). Correct answers are highlighted in bold.**

[Figure]

[Figure]

**Figure 9: results questionnaire for specific questions Q16b-Q20b for a) the class who participated in the lesson (4C) and b) the**

**classes who did not participate in the lesson (4A-4B-4D). Correct answers are highlighted in bold.**

[Figure]

**4.2 General audience questionnaires**

Twenty-six participants completed both pre- and post-experiment questionnaires. The pre-experiment questionnaire indicates that the participants already had a solid understanding of the tsunami phenomenon, as reflected in questions Q2i-Q6i (Fig.10). In particular, Q2i-3i show that most of the participants were aware of the potential tsunami risks of their country or of the locations they visited; Q4i-Q5i demonstrate that they understood both the processes behind tsunami generation and the potential impacts on people; Q6i provides a reassuring insight into the confidence participants place in scientific investigation. In contrast, given that the majority of participants live >3 km away from the coast (Q1i), Q2i shows that their perception of the tsunami hazard in their own country is less developed. This likely reflects the fact that, as all respondents were Italian, they did not perceive Italy as a country at high risk for tsunamis, thereby underestimating its potential impact.

Concerning the post-experiment questionnaire, in addition to the general appropriateness of both the participants' competences in understanding what is shown in the experiments (Q7i) and the language used by the demonstrators (Q8i), the experiment stimulated the interest in research of most of the respondents (Q9i). This is further supported by the results in Q10i-Q12i, where the majority of respondents answered correctly:

- for Q10i, > 80% of the respondents answered correctly.
- for Q11i, > 40% of the respondents answered correctly.
- for Q12i, > 75% of the respondents answered correctly.

We highlight that Q12i testifies that the majority of the participants gained fundamental knowledge on tsunami hazard in Italy as they did not have a preliminary perception of this (see Q2i).

[revised manuscript text omitted]

---

## Author Response (AR1)

**Dear editor,**

We report in this document a point-by-point answer to the questions raised by the two reviewers. Unless differently stated, line numbers refer to the manuscript with tracked changes. In addition to the changes reported in the following pages, we have also slightly modified Fig. 3 and edited some sparse words that do not alter the scientific content of the paper.

**Reviewer #1 (Anonymous)**

This manuscript explores various routes toward quantitative teaching tools for Earth Sciences and for certain hazardous phenomena in particular. The article is well laid out, the introduction is engaging and interesting such that it draws the reader into the rest of the work, and the findings are clear, well presented, and useful. I commend the authors on this work and look forward to citing it in future work.

We thank the reviewer for these nice comments on our manuscript. In the following we record our actions to answer the raised issues.

**A few points to consider**

-- Line 33-34. "cannot provide insights on the physical mechanisms which cause such spectacular manifestations" -- Why not? I could imagine some cases where the museum approach would yield some sense of insight for the viewer. Could the authors support this claim a little more?

What we wanted to convey with this statement is that the splendid collections of minerals and rocks allow pure, sometimes even in-depth, observations. Although they are often accompanied by illustrative panels with schematic explanations of the formation processes of the observed rocks, the lack of direct experience through the act of "measurement" can make it more difficult to understand the dynamics and evolution of natural phenomena over time. However, we realize that our sentence can lead to some misunderstanding with respect to the importance of the classic approach in outreach initiatives. We have therefore reformulated LINES 36-40 of the original manuscript as follows:

Although rock samples and field activities represent the basis of the geological disciplines and natural phenomena can be explained from a purely theoretical perspective, learning is more effective when a descriptive approach is coupled with practical experiments or unconventional techniques. (LINES 37-41).

-- Line 176. The equation given is for a solid particle moving through a viscous liquid. For a spherical bubble, instead of 2/9 it is 1/3. That can be found in the excellent book "Bubbles, drops, and particles" by Grace, Weber, and Clift. And on line 179 there is a typographic error with 10^-3 instead of 10-3.

Thank you for this observation. We have corrected the typo on line 179. Regarding the equation, we have used equation 3-18 in Chapter 3, p. 35 from:

Bubbles, Drops, and Particles by Clift, Grace, Weber, ACADEMIC PRESS New York San Francisco London 1978.

In this equation (valid for rigid spheres) it is considered 2/9 instead of  $\frac{1}{3}$ . We have also added the reference to this book in the text (LINE 182).

-- Check references carefully (note difference between "Illingworth" and "Illingsworth" throughout).

Thank you, we have modified the reference throughout the paper (LINE 404).

- -- I wonder what the authors think about the fundamental limitations of not using real magmas to teach and inspire people concerning volcanic eruptions? Are simplistic demonstrations too simplified? Or too abstracted from reality? See a brief discussion of how transformative it can be to use real magmas or real high temperature silicate systems here: https://doi.org/10.1038/s41561-018-0283-5
- -- Have the authors seen the paper about lava flow dynamics using fudge? I mention this because it has similar learning goals to the bubble rise activity explored herein.

**https://doi.org/10.5408/1089-9995-56.1.73**

Thanks for these inspiring comments! Using real magma can be useful in many outreach initiatives for simulating lava flows, bubbling lava lakes, or exploring magma's interaction with rocks etc... However, using real magma requires expensive equipment capable of maintaining temperatures above 900°C (at least), and generally above 1100°C for basaltic magma compositions. These instruments generally cannot be brought to schools or public places. Moreover, in the proposed experiment, we would like to quantify the process of bubble rise within a magma column, and to do so, we need a transparent material that allows us to observe and measure the size and rate of ascent of the bubbles.

We also thank the Reviewer for bringing these published works to our attention, which we have now referred to in our revised manuscript (LINES 157-160):

Some experiments used molten real magma or silica glasses at temperatures  $\geq 900-1\,100\,^{\circ}\mathrm{C}$  to illustrate the dynamics of volcanic processes at a general public (Wadsworth et al., 2019). In the context of our purpose, we used an analogue, low temperature material (see also Rust et al., 2008), which is easily and safely transported outside the laboratory, allowing us to quantify the physical properties being transparent.

-- A key issue in teaching Earth Sciences is the subject of scaling. How do we scale from one large phenomenon in Earth Science, to something small enough that we can deal with it at the laboratory or classroom scale? The answer is typically to use dimensional analysis. I think some discussion of this could be good - including how rich it can make classroom demonstrations if that problem of scale is taught and explained. What do the authors think of that?

We totally agree with this comment: to fully understand natural phenomena, it is essential that laboratory measurements will be scaled to the real case. This topic should be addressed in classrooms, and the comparisons between measured quantities in lab with the natural one should be the starting point for understanding the problem.

We have added the following sentence at LINES 55-57:

Adapting the wide range of natural phenomena to the laboratory environment requires a dimensional scaling of the measured quantities (Merle, 2015). In order to become more effective, this aspect needs to be properly addressed during outreach activities.

**Reviewer #2 (Anonymous)**

We thank the reviewer for his/her insightful comments. In the following we record our actions that answer the raised issues. Unless differently stated, line numbers refer to the manuscript with tracked changes.

-- Lines 117 - 118: How do you know that the target audience actually benefitted? Perhaps rephrase this to say- to evaluate differences in perception of the two audience groups.

We agree that in the present form the text could be misleading, and we therefore modified the text accordingly (LINES 122-124):

This allowed us to evaluate the perception of the two audience groups with respect to the same type of information.

-- Lines 309 - 311: Break this down based on the categories being compared i.e. from 4C (what %) and what % from 4A-4B-4D. This is a bit confusing

We modified the text as follow (LINES 317-322):

Comparing the answers on general topics from the two groups of students (Fig. 6), we found that most students reported slight to moderate interest in scientific research: in group 4C, 20% were slightly interested, ~67% moderately interested, and ~13% very interested; in groups 4A–4B–4D, 34 % were slightly interested, ~49% moderately interested, and ~11% very interested (Q1b).

-- Line 325: Overall, the numerical question numbers by themselves, don't make any sense. It would be more effective to compare, for example, 73% of these students correctly answered "what factors influence the type of an eruption" (Q11b). Write this for all the questions for the ease of your reader.

We modified the text in order to report for each question ID, the full or a short version of the question text (LINES 337-350).

Approximately 73% of these students answered correctly to the question "Which factors influence the type of an eruption?" (Q11b), in comparison to ~66% of students from the other classes, while both groups achieved similar success on the question "What is the role of viscosity in explosive eruptions?" (Q12b) with ~73% correct responses. For Q13b (relationship between magma viscosity and rising speed of gas), ~80% of class 4C answered correctly, versus ~64% in classes 4A-B-D, and for Q15b (importance of combining numerical and laboratory experiments), ~87% of class 4C responded correctly compared to  $\sim$ 83% in the other groups. However, Q14b (calculation of the viscosity of a liquid) was answered more accurately by students from classes 4A-B-D (~85% correct) than by those from class 4C (~73% correct). Regarding seismological-related questions (Q16b-Q20b), the data reveal some interesting trends. For Q16b (earthquake magnitude increase from M5 to M3), students in class 4C performed better with ~47% correct answers compared to ~28% in the other groups, whereas for Q17b (find the correct answer about seismic waves) and Q18b ("What are dromocrones?") both groups obtained similar results, with correct responses rates of ~80% on each question. Q19b ("What does a seismometer measures?"), however, was a challenge for both groups, with only ~20% of class 4C and ~28% of classes 4A-B-D answering correctly. Finally, for Q20b (differences between Mercalli scale and Richter scale), students from classes 4A-B-D provided a higher percentage of correct answers (~60%) than those from class 4C (~40%).

-- Line 354: again, say that most of the participants were aware of the potential tsunami risks of their country or the locations they visited (Q2i-Q3i).

Thanks, we modified the text as suggested (LINES 366-373):

The pre-experiment questionnaire indicates that the participants already had a good understanding of the tsunami phenomenon, as reflected in questions Q2i-Q6i (Fig.10). In particular, most of the participants were aware of the potential tsunami risks of the locations they visited (~69% for Q3i), they understood both the processes behind tsunami generation and the potential impacts on people (~73% for Q4i - ~88% for Q5i), and they had a reassuring confidence in scientific investigation (~70% - Q6i). In contrast, given that the majority of participants live >3 km away from the coast (~85% - Q1i), their perception of the tsunami hazard in their own country is less developed (~73% of them answered that their country is not at tsunami risk - Q2i).

-- Lines 364-370: Again, here, please summarize what you are asking and then elaborate on what it can indicate in terms of effectiveness of the outreach.

Thanks, we modified the text as suggested (LINES 379-390):

This is further supported by the results in Q10i-Q12i, where the majority of respondents answered correctly:

- for Q10i ("On average, at what speed do tsunami waves travel at Stromboli?"), > 80% of the respondents answered correctly.
- for Q11i ("What information can we directly obtain from the study of waveforms?"), > 40% of the respondents answered correctly.
- for Q12i ("Which geographical areas of Italy do you think are most likely to be affected by volcanically-induced tsunamis?"), > 75% of the respondents answered correctly.

We highlight that responses to Q10i-Q11i demonstrate that even complex and quantitative scientific contents were well received, while Q12i testifies that most participants gained fundamental knowledge about tsunami hazard in Italy, information they lacked beforehand (see Q2i).

---

## Author Response (AR2)

THANKS FOR THE EDITORIAL COMMENTS, WE HAVE DONE WHAT SUGGESTED (SEE FOLLOWING LINES. IN ADDITION, WE HAVE ALSO CHANGED FEW TYPOS.

Please link a copy of the full questionnaire in section 3.2 of the Methods section in the text.

DONE (LINES 300-301). WE HAVE INCLUDED THE QUESTIONNAIRES (AS SUPPLEMENTARY MATERIAL) IN BOTH ITALIAN AND ENGLISH.

Additional private note (visible to authors and reviewers only): Is it a mistake on Line 339. Magnitude increase? Or decrease?

THANKS, WE HAVE CORRECTED LINE 340.

Please double check all questions are correctly noted.

DONE.

Please also reference in the text and please link a copy of the full questionnaire in section 3.2 of the Methods section in the text.

DONE (LINES 300-301). WE HAVE INCLUDED THE QUESTIONNAIRES (AS SUPPLEMENTARY MATERIAL) IN BOTH ITALIAN AND ENGLISH.